# Love on Empty: The Development and Validation of a Comprehensive Scale to Measure Burnout in Modern Relationships

**DOI:** 10.3390/bs15121737

**Published:** 2025-12-16

**Authors:** Ashley Elizabeth Thompson, Ryn Theis, Rachel Willhite, Julitta Dębska

**Affiliations:** 1Department of Psychology, University of Minnesota Duluth, 347B BohH 1207 Ordean Court, Duluth, MN 55812, USA; theis676@umn.edu (R.T.);; 2Doctoral School of Social Sciences, Univeristy of Łódź, Narutowicza 68, 90-136 Łódź, Poland; julitta.debska@edu.uni.lodz.pl

**Keywords:** relationship burnout, scale development and validation, relational overload and depletion, relationship satisfaction

## Abstract

Modern romantic relationships face increasing internal and external pressures that may leave partners emotionally depleted and overwhelmed, yet empirical tools for assessing relationship burnout remain limited, mononormative, and psychometrically underdeveloped. Across two studies, we developed and validated the Antecedents of Relationship Burnout Scale (ARBS), a multidimensional measure grounded in the Job Demands–Resources (JD–R) model and designed to capture the relational demands and resource deficits that precipitate burnout. Study 1 generated and evaluated an initial 51-item pool using a sample of 175 partnered adults. Exploratory factor analysis revealed a clear and robust two-factor structure: Relationship Depletion and Exhaustion (e.g., emotional detachment, diminished appreciation, unmet emotional/sexual needs) and Relational Overload (e.g., external stressors, partner demands, role strain). Study 2 sought to confirm this structure and establish the ARBS’s psychometric validity via a sample of 288 adults. A confirmatory factor analysis supported a 36-item two-factor model with strong internal consistency, full measurement invariance across gender, and theory-consistent associations with relationship satisfaction, therapy participation, and infidelity urge, demonstrating both convergent and predictive validity. Together, these studies introduce the ARBS as the first comprehensive, theoretically grounded measure of the antecedents of relationship burnout, offering a rigorous foundation for future research, assessment, and intervention.

## 1. Introduction

Scholars repeatedly uncover evidence indicating that romantic relationships offer a variety of benefits, including emotional support, companionship, a sense of belonging, reduced stress levels, mental/physical health, and more ([11]; [30]; [68]). That said, these benefits result when one’s relationship flourishes. Conversely, struggling relationships can have the opposite effect, resulting in a reduction in happiness, increased risk for anxiety/depression, and physical/mental health concerns ([63]; [72]).

Recent research suggests that romantic partners from younger generations may be experiencing greater struggles than those from older generations due to increased pressure and expectations in romantic relationships ([3]; [25]). Indeed, [24] ([24]) argues that adults are placing more pressure than ever on their romantic partner to fulfill needs associated with self-esteem, self-expression, and personal growth. This increased emphasis on relationship processes that require mutual insight means that investing time and energy in one’s relationship is much more important today than in the past. As a result, romantic partners from younger generations are required to invest more resources to meet their partner’s expectations than those from older generations.

Beyond these intensified internal demands, couples today also navigate a growing number of external stressors that can undermine relational stability and satisfaction ([18]). Factors such as work overload, financial uncertainty, and global or societal crises have been shown to increase tension between partners and reduce emotional resources ([79]; [76]; [36]). These contextual pressures often spill over into the relationship, disrupting emotional connection and communication ([58]; [51]). According to the Vulnerability–Stress–Adaptation framework ([34]), such stressors interact with individual vulnerabilities and coping processes, influencing how partners adapt to ongoing demands. Together, these internal expectations and external pressures create a relational climate in which sustaining mutual well-being requires continual emotional and cognitive investment.

The increased investment and stress burden in modern-day romantic relationships parallels the heightened demands placed on younger generations in the workplace ([55]). In fact, the modern work environment is often described as comprising more digitized, high-intensity work characterized by constant connectivity, blurred boundaries between work and private life, and limited opportunities for recovery ([13]; [41]; [43]). This 24/7 availability culture leads to cognitive overload and emotional strain, resulting in a greater and more sustained investment of time and energy ([42]; [66]; [83]). This large amount of effort, coupled with high performance demands, can result in burnout, a term first coined by Herbert Freudenberger to refer to high ideals and severe stress in caregiving professions ([26]). More recently, via the ICD-11, burnout has been defined as chronic workplace stress that has not been successfully managed and is characterized by three dimensions: exhaustion, mental distancing from one’s job, and reduced efficacy ([80]).

Building on the construct of workplace burnout, it is plausible that the substantial emotional and cognitive investment in modern romantic relationships, and the accompanying stress, may also lead to a form of burnout (i.e., relationship burnout; RB). Although there has been relatively little empirical attention on RB, the few studies that exist have referred to the phenomenon as “couple burnout”, described as emotional, physical, and cognitive exhaustion experienced by romantic partners trying to attain unrealistic expectations within a relationship ([2]). Most studies (but not all) have focused on the role of infertility and infidelity in the development of couple burnout, revealing that navigating these challenges can lead to declines in intimacy, communication, and overall relationship satisfaction ([4]; [5]; [27]; [52]; [55]; [75]). Taken together, these findings underscore a significant research gap and point to the need for systematic investigation of relationship burnout as a distinct phenomenon within the broader literature on romantic relationships.

### 1.1. Limitations of Existing Research

While this emerging literature has shed light on an important and underexplored area, it remains narrow in both scope and framing. The term “couple burnout” is inherently mononormative, rooted in the societal assumption that romantic relationships consist of two people only ([60] [48]). Indeed, in Western cultures, monogamy is often taken for granted as the default and privileged relational structure, which shapes the norms, expectations, and even research agendas surrounding intimate partnerships ([29]; [67]). Scholars have increasingly called for research that challenges mononormativity by broadening our conceptualization of relationships beyond monogamous dyads ([48]). In response to these calls and to better reflect relational diversity, the present study adopts the more inclusive term “relationship burnout” (RB).

Beyond terminological concerns, methodological limitations further constrain our understanding of burnout in relationships. The only existing scale measuring burnout in the context of romantic relationships has undergone little or no psychometric evaluation. In fact, [54] ([54]) created the unidimensional “Couple Burnout Scale”) by modifying the General Burnout Measure ([53]). Although the original burnout items could describe RB, limited scientific rigor employed when developing the Couple Burnout Scale raises concerns regarding its validity. In particular, the items themselves (e.g., “weak/sickly,” “difficulties sleeping,” “I’ve had it”) are overly vague and may capture exhaustion or distress stemming from sources unrelated to one’s romantic relationship. Moreover, the scale focuses exclusively on assessing the affective/psychological experience of burnout (i.e., symptoms), rather than accounting for the relational processes and contextual factors that may cause burnout to emerge. This narrow focus is problematic given that burnout researchers emphasize the importance of examining both symptoms and antecedents, as causes provide a more precise understanding of how burnout originates and how it can be prevented ([22]; [64]; [82]). For example, in the occupational domain, burnout is understood as arising from two broad domains: individual factors (e.g., personality or coping strategies) and organizational factors (e.g., workload, fairness, demands; [39]). By extension, a measure of relationship burnout that does not attend to causal factors (e.g., inequities in effort, unmet expectations, lack of support) offers only a partial picture. Thus, a multidimensional assessment that captures the antecedents of RB is essential for advancing conceptual clarity and intervention development.

### 1.2. Theoretical Framework

To better understand the multidimensional nature of relationship burnout, the Job Demands–Resources (JD–R) model ([6]; [16]) provides a particularly strong framework. Extensively validated in occupational contexts, the JD–R model demonstrates that burnout emerges when chronic demands outweigh available resources, leading to exhaustion and disengagement. This logic translates seamlessly to intimate partnerships, where relational demands (e.g., conflict, inequities, role strain) and relational resources (e.g., support, communication, responsiveness) are critical determinants of relationship quality and well-being ([34]; [61]).

Indeed, research has revealed that higher relational demands are associated with increased distress and decreased satisfaction (e.g., [17]; [69]; [59]; [49]; [32]), whereas greater relational resources such as dyadic coping, responsiveness, and emotional support predict higher satisfaction and greater resilience (e.g., [14]; [34]; [56]).

Moreover, the Job Demands–Resources (JD–R) framework has been successfully adapted to other life domains, including parental burnout (via the Balance between Risks and Resources model; [47]), academic settings (via the Study Demands–Resources framework; [40]), and the interface between work and family life (via the Work–Home Resources model; [71]). These adaptations demonstrate the JD–R model’s versatility in explaining stress, exhaustion, and engagement across diverse non-occupational contexts. Building on this theoretical foundation, the present research extends the JD–R model to romantic relationships, conceptualizing RB as a multidimensional construct shaped by the dynamic interplay between relational demands and resources.

### 1.3. The Current Research

While the JD–R model provides a strong conceptual rationale for understanding how relational demands and resources may give rise to relationship burnout, empirical research has yet to capture this complexity in a reliable, validated measure. Existing approaches, most notably the Couple Burnout Scale ([54]), are limited by vague item content, a unidimensional structure, and a narrow focus on affective symptoms. Consequently, there is a pressing need for a psychometrically rigorous tool that reflects the multidimensional nature of RB and accounts for the contextual factors highlighted by the JD–R framework.

Thus, the current research comprises two studies designed to address key limitations in the existing literature on RB by developing and validating a psychometrically sound, multidimensional measure of the construct. Guided by the JD–R framework ([6]; [16]), we propose that RB reflects the dynamic interplay of relational demands (e.g., conflict, inequities, unmet expectations) and relational resources (e.g., support, responsiveness, communication). Accordingly, we expected a two-factor structure to emerge that parallels the JD–R model, capturing both the strain created by excessive demands and the protective role of relationship resources.

With this in mind, Study 1 sought to explore the factor structure by examining whether items clustered meaningfully around relational demands and relational resources. Study 2 aimed to confirm the factor structure identified in Study 1 and to assess the psychometric properties of the new scale, including reliability and construct validity. Importantly, these studies adopted inclusive language to avoid mononormative assumptions, recognizing that not all relationships are dyadic, in line with recent calls for more inclusive relationship science (e.g., [48]). By grounding measurement development in the JD–R framework, this research provides a theoretically informed foundation for future work examining the predictors, outcomes, and potential interventions related to RB across diverse relational structures.

## 2. Study One

### 2.1. Method

#### 2.1.1. Participants

A total of 200 participants (all of whom were required to be in a relationship for at least six months) were recruited to participate in Study One. Of these, 13 failed to complete the study in its entirety, nine failed one of two attention check items, and another three completed the study in under two minutes. Thus, our final sample comprised 175 adults (53.7% women, 43.4% men, 2.9% gender diverse) with a mean age of 38.01 years (*SD* = 12.65). A total of 63.4% of the participants identified as being White, 31.6% as Black/African American, 2.9% as Asian, 1.1% as American Indian, and the remaining 1.0% identified as “other” or “from multiple races.” Additionally, 78.1% of the participants identified as heterosexual, 15.6% as bisexual, 5.1% as lesbian, 1.1% as gay, and the remaining 0.2% as pansexual or “other.” The majority of participants reported being married (69.8%), and 30.2% in an “unmarried” monogamous relationship. When asked how committed they were to their primary partner on a 7-point scale from (not at all committed to very committed), respondents indicated an average score of 6.25 (*SD* = 1.08).

#### 2.1.2. Measures

Demographics Questionnaire. Participants also completed a demographics questionnaire including items to assess age, gender identity, race, sexual identity, and relationship status.

Antecedents for Relationship Burnout Scale (ARBS). Items for the ARBS were generated using several online forums (e.g., Reddit, Quora, Answers.com) and social media platforms. Respondents to these prompts were asked to describe “what relationship burnout means, whether they had ever experienced it, and what may have caused it.” Over the course of two weeks, 102 internet users provided responses, offering a wide range of descriptions and explanations. The first and second authors then conducted a thematic analysis of these responses, which informed the initial pool of items for the ARBS.

The preliminary draft of the ARBS contained 51 items and was piloted with a sample of 20 undergraduate students. Undergraduate research assistants were recruited from various research labs at the P.I.’s institution. To be eligible, students were required to be at least 18 years old and have prior experience with romantic relationships. To maximize efficiency and reduce participant burden, no additional demographic information was collected from this group. The purpose of this pilot study was to ensure that key antecedents of relationship burnout had not been overlooked, that item wording was clear, and to aid in refining the instructions and response scale. Based on pilot feedback, two additional items were created, resulting in a first full draft of 51 items. All items were rated on a 7-point Likert scale ranging from 1 (not at all) to 7 (all the time). The final instructions read: “Below you will find a series of statements that people commonly use to explain why they may feel ‘burnt out’ in their relationship from time to time. With these in mind, please use the following 7-point scale to describe the extent to which each statement describes your current romantic relationship in the past 6 months.”

### 2.2. Procedure

Study One was conducted in accordance with the principles outlined in the Declaration of Helsinki. Ethical approval was obtained from Institutional Review Board at The University of Minnesota (STUDY00025158), ensuring that our study adhered to both national and international guidelines. After obtaining IRB approval, participants were actively recruited via Prolific^®^. All participants were informed that they were participating in a 30 min anonymous survey on understanding romantic relationships. Interested participants were automatically redirected to Qualtrics^®^ and asked to review an informed consent form, which provided more information about the study. Consenting participants were then asked to complete the newly formed ARBS and demographic information (along with measures unrelated to Study One), all hosted via Qualtrics^®^. See our OSF folder for all measures (https://osf.io/9cner/overview?view_only=9879244dbf8f4a4b8bd6ad3d33d3990b, accessed on 14 November 2025). After the completion of the measures, participants were debriefed and compensated with a $4.00 deposit into their Prolific account.

### 2.3. Results

All raw data is anonymized and housed on our OSF page (https://osf.io/9cner/overview?view_only=9879244dbf8f4a4b8bd6ad3d33d3990b). These data were screened and cleaned according to procedures outlined by [70] ([70]). Although no outliers were identified, all the items demonstrated significant skew (evidenced by a skew *z* score of 2.58 or higher). Thus, an exploratory factor analysis with principal axis factoring extraction and a Promax rotation was adopted because of the numerous normality violations. The results from the KMO test (0.95) revealed that the sampling adequacy was appropriate for an EFA. Additionally, Bartlett’s test of sphericity was used to analyze the adequacy of correlations between items and was found highly significant, *χ*^2^(1275) = 8763.11; *p* < 0.001.

After determining that an EFA was appropriate, the results revealed nine factors with eigenvalues greater than one, but the scree plot and a parallel analysis indicated that a two-factor solution was best. Thus, a second principal axis factoring EFA with a Promax rotation that restricted the model to two factors was conducted. Scale items were retained if they had a factor loading of 0.35 or above on one of the factors but no cross-loadings (i.e., greater than 0.35 on two or more factors). Although all items loaded on at least one factor, two loaded on more than one factor (“My partner does not support my decisions” and “My financial needs are not being met”). Additionally, one item had a factor loading of 1.01 (“I feel detached from my partner”). To explore this concern, inter-item correlations were conducted. This item was correlated with “I no longer feel valued by my partner” at *r* = 0.89. Thus, “I no longer feel valued by my partner” and the two cross-loaded items were removed, and a final EFA was conducted with the remaining 48 items.

Using the revised set of items, the final principal axis factoring EFA with a Promax rotation was conducted. The KMO test result was 0.95 and Bartlett’s test of sphericity was significant, *χ*^2^(1128) = 8017.86, *p* < 0.001. The two resulting factors were then interpreted using the JD–R framework that guided the current research (see Table 1 for final factor loadings and Table 2 for the means and standard deviations for all items). The first factor, Relationship Depletion and Exhaustion (accounting for 55.91% of the variance), comprised 32 items reflecting detachment, loss of compatibility, diminished appreciation, lack of communication, and unmet emotional/sexual needs (e.g., “I feel detached from my partner,” “We do not communicate enough,” “My emotional needs are not being met”). These items capture the gradual depletion of affective resources and withdrawal of positive engagement within the relationship. Consistent with frameworks of burnout, this pattern aligns with the concept of emotional exhaustion and was therefore labeled the Relationship Depletion and Exhaustion subscale. Cronbach’s alpha indicated that the Relationship Depletion and Exhaustion subscale demonstrated excellent internal consistency (*α* = 0.98).

The second factor, Relational Overload (accounting for 5.30% of the variance), consisted of 16 items describing external and systemic stressors (e.g., family tension, competing responsibilities, prior relationship trauma) as well as overextension within the dyad (e.g., “My partner requires too much reassurance,” “I do not have enough personal space”). These items reflect the accumulation of demands and pressures that extend beyond relational dynamics themselves, taxing individuals’ capacity to invest in their partnership. In line with the JD–R model ([6]; [16]), which emphasizes burnout as the result of excessive demands relative to available resources. The Relational Overload subscale also demonstrated great internal consistency (*α* = 0.94).

Finally, to examine the correlation between the Relationship Depletion and Exhaustion and Relationship Overload subscales, a Pearson Product-Moment Correlation Coefficient was conducted. The results indicated that the two factors were strongly correlated, *r* = 0.83, *p* < 0.001, indicating substantial overlap between the two subscales. Although one clear instance of item redundancy was addressed prior to the final EFA, the high correlations among subscales and several exceptionally strong factor loadings suggest that additional overlap may remain. Nevertheless, we proceeded with a CFA in Study 2 to more rigorously evaluate the stability and interpretability of the emerging factor structure. Prior work supports using CFA even when EFA results indicate potential item overlap, as CFA offers a stricter test of model fit, enables comparison of competing structures, and allows examination of theoretically informed constraints not testable in EFA (e.g., [10]; [81]). Thus, advancing to CFA was methodologically and theoretically justified to determine whether the structure demonstrated adequate fit and construct validity.

### 2.4. Discussion

The purpose of Study One was to provide an initial test of the proposed structure of the Antecedents for Relationship Burnout Scale (ARBS). Although the JD–R framework ([6]; [16]) provided a useful starting point for conceptualizing relationship burnout, the current results suggest that its application to intimate partnerships may be somewhat limited. Consistent with the model, our two-factor solution revealed two dimensions that can broadly be mapped onto demands and resources. Specifically, the Relational Overload subscale parallels the JD–R notion of excessive demands, capturing external pressures and partner expectations that tax individuals’ capacities. Likewise, the Relationship Depletion and Exhaustion subscale resembles the erosion of resources, as it reflects the absence of appreciation, intimacy, and support that would otherwise buffer stress. These parallels highlight the value of the JD–R model in framing relationship burnout as a dynamic imbalance between demands and resources and add to a growing body of research demonstrating that the JD–R has been successfully extended to other life domains (e.g., [47]).

However, based on the initial results of the EFA, the JD–R model appears not to fully capture the nuanced ways burnout emerges in relational contexts. First, items included in the Relationship Depletion and Exhaustion subscale extend beyond a simple “resource deficit” and reflect a broader process of disengagement, detachment, and emotional withdrawal that is more consistent with the exhaustion and depersonalization dimensions of traditional burnout models (e.g., [46]). Second, some of the items in the Relational Overload subscale (e.g., trauma histories, infertility, family strain) represent contextual stressors that are not easily classified as either “demands” or “resources” in the JD–R sense. In sum, the results from Study One provide some initial evidence that although the JD–R model is a useful heuristic for understanding the antecedents of relationship burnout, it may not fully account for the complexity of relational dynamics.

Nevertheless, two important components of relationship burnout emerged from the analyses. The first factor, Relationship Depletion and Exhaustion, represents the erosion of affective and psychological engagement within a partnership. Items in this factor describe emotional detachment, diminished appreciation, boredom, poor communication, resentment, and unmet emotional or sexual needs. These patterns are consistent with research showing that chronic relational conflict, stress, disengagement, and unmet expectations are among the strongest predictors of relationship distress and dissolution ([17]; [57]; [74]). This factor highlights how the loss of connection, fun, and appreciation undermines satisfaction and can place partnerships on a trajectory toward dissolution.

The second factor of the ARBS, Relational Overload, reflects the weight of excessive external demands and systemic pressures. Items in this factor highlight family strain, competing responsibilities, trauma histories, loss of autonomy, etc. This dimension aligns with the literature demonstrating that external stressors (e.g., family tension and financial strain) can spill over into romantic relationships, eroding quality and increasing the risk of dissolution ([34]; [50]; [51]). These findings underscore how relational overload functions as a critical antecedent to burnout: when external pressures accumulate without sufficient buffering, couples face a heightened likelihood of distress and disengagement (i.e., burnout).

Finally, the two subscales for the ARBS were strongly correlated. Although this high correlation suggests a degree of redundancy, such an association is theoretically consistent with research showing that elevated relational demands deplete partners’ emotional resources, producing exhaustion and reduced capacity for positive engagement ([58]; [59]). Dyadic stress processes also “crossover” within couples such that one partner’s demands and strain increase the other’s exhaustion, amplifying a mutually reinforcing cycle of overload and depletion ([7]). In sum, the two dimensions of RB are closely intertwined, reflecting the reciprocal and mutually reinforcing nature of relational demands and emotional exhaustion within romantic partnerships.

## 3. Study Two

Although Study One provided strong initial evidence for a two-factor structure of the ARBS, additional work is needed to establish the robustness and validity of this measure. Specifically, a confirmatory factor analysis (CFA) is required to formally test whether the proposed two-factor model offers a good fit to the data. In addition, it is essential to reassess the internal consistency of the resulting subscales and to evaluate the scale’s construct validity. To this end, Study Two was designed to (a) confirm the factor structure identified in Study One using CFA, (b) examine the reliability of the subscales, and (c) explore convergent and predictive validity by assessing associations between ARBS scores and three theoretically relevant outcomes: relationship satisfaction, prior engagement in relationship therapy with one’s current partner, and one’s urge to commit infidelity.

Relationship satisfaction is one of the most robust indicators of relational functioning and well-being, and decades of research indicate that heightened conflict, inequities, and lack of support are consistently associated with reduced satisfaction (e.g., [9]; [15]; [23]). Thus, higher scores on both ARBS subscales are expected to be associated with lower levels of satisfaction. In addition, engagement in relationship therapy is typically sought when partners perceive significant difficulties, unmet needs, or persistent distress ([38]; [37]; [19]). In fact, previous research demonstrates that romantic partnerships experiencing diminished intimacy, poor communication, and high levels of strain are more likely to pursue and sustain therapy as compared to those not experiencing these concerns ([20]; [8]). Finally, recent research has indicated that interpersonal factors (e.g., relationship satisfaction, love, desire) are the most significant predictors of infidelity ([77]; [78]). Thus, scores on the two subscales of the ARBS were expected to significantly predict therapy participation and the urge to engage in infidelity.

### 3.1. Participants

A total of 300 U.S. residents were recruited to participate in Study Two, all of whom were required to have been in a relationship with their current romantic partner for at least six months. Following data screening, seven failed the attention check item, three provided an IP address outside of the U.S., and two completed the study in under two minutes. Thus, our final sample comprised 288 adults (59.4% women, 37.2% men, 3.4% gender diverse) with a mean age of 41.72 years (*SD* = 12.10). A total of 77.9% of the participants identified as being White, 9.0% as Asian, 8.7% as African American, 2.4% as Native American, and the remaining 2.0% identified as “other” or “from multiple races.” Additionally, 77.4% of participants identified as heterosexual, 13.5% as bisexual, 3.1% as lesbian, 2.1% as gay, 3.1% as pansexual, and the remaining 0.8% as “other.” Finally, 60.5% of participants reported currently being married, 38.5% in a monogamous relationship, and the remaining 1.0% in an open/polyamorous relationship. When asked how committed they were to their primary partner, respondents indicated an average score of 6.55 (*SD* = 1.02).

### 3.2. Measures

Demographics Questionnaire. Participants completed a nearly identical demographics questionnaire as compared to the version used in Study One. However, in Study Two, a dichotomous item assessing previous participation in relationship therapy was included. Participants were asked, “Have you ever sought therapy, counseling, or relationship support due to relationship stress?”, in which “yes” and “no” were response options. An item assessing one’s urge to engage in infidelity was assessed via the following item: “To what extent do you currently feel the urge to act unfaithfully in your relationship (i.e., engage in emotional, romantic, and/or sexual behaviors with someone other than your current romantic partner)?”. This item was then rated using a 5-point scale from 1 (not at all) to 5 (a lot).

Antecedents for Relationship Burnout Scale (ARBS). The 48-item ARBS that was developed in Study One was adopted in Study Two.

Relationship Assessment Scale (RAS; [31]). The RAS is a 7-item self-report questionnaire developed to assess global relationship satisfaction. The RAS has been validated across diverse populations, including married couples, dating couples, cohabiting partners, and individuals involved in various intimate relationships ([1]). The scale uses a 5-point Likert-type response scale, with “1” indicating a low level of satisfaction and “5” reflecting a high level of satisfaction. Sample items include “How good is your relationship compared to most?” and “How often do you wish you hadn’t gotten in this relationship?”. The RAS demonstrated excellent internal consistency in Study Two as evidenced by a Cronbach’s alpha of 0.93.

### 3.3. Procedure

Once again, IRB approval was obtained from the relevant institutional review board (STUDY00025158), and participants were once again collected via Prolific^®^ using a recruitment message indicating that we were recruiting U.S. adults for a 20 min survey on factors impacting romantic relationships. Interested participants were automatically directed to an informed consent form, which provided more information about the study and notified them that they needed to be in a relationship with their current partner for at least 6 months. Consenting participants were then asked to complete the revised ARBS, the relationship satisfaction scale, and demographic information (including the item assessing therapy participation). After the completion of the measures (which can be viewed on our OSF page: https://osf.io/9cner/overview?view_only=9879244dbf8f4a4b8bd6ad3d33d3990b), participants were debriefed and compensated with $3.00.

### 3.4. Data Preparation and Cleaning

Again, data are housed on our OSF page (https://osf.io/9cner/overview?view_only=9879244dbf8f4a4b8bd6ad3d33d3990b) and were cleaned according to procedures outlined by [70] ([70]). Across the sample, there were 89 participants who were missing at least one item on the ARBS. Additionally, like Study One, all the items demonstrated significant skew. Thus, we estimated confirmatory factor models using robust maximum likelihood (MLR) with full information maximum likelihood (FIML) for missing data. Model selection followed established recommendations by prioritizing changes in fit (ΔCFI, ΔRMSEA) and robust Satorra-Bentler *χ*^2^ difference tests when models were nested.

### 3.5. Results

#### 3.5.1. Confirmatory Factor Analysis

The results of the CFA indicated that the proposed model had less-than-adequate fit (as evidenced by the model fit indices; RMSEA  =  0.10, CFI  =  0.71, TLI  =  0.70). To improve model fit, items loading under 0.60 on a given factor were omitted. In addition, overlapping items were identified via modification indices. Overall, 12 items were omitted because they failed to load on their associated factor, and two were removed because they were implicated in many of the modification indices. Improvements to the model’s fit were made by allowing for covariance between items belonging to the same factor.

After removing items contributing to poor fit and allowing for covariance within factors, the final CFA was conducted (see Figure 1). All the model fit indices suggested adequate fit, RMSEA = 0.08 (90% CI [0.078–0.084]); CFI = 0.89; TLI = 0.89. Although slightly below the conventional 0.90 threshold, these indices are considered acceptable for models with complex, multidimensional constructs ([44]; [65]). Standardized factor loadings ranged from 0.60 to 0.88, suggesting good convergent validity among the items. Thus, the final version of the ARBS comprised 36 items organized into two subscales: the Relationship Depletion and Exhaustion subscale (containing 26 items) and the Relational Overload subscale (including 10 items). The item mean scores ranged from 1.66 to 2.98, indicating a low level of agreement with the items. See Table 2 for means and *SD*s and Appendix A for final scale.

To explore competing models, we compared the two-factor structure to a single-factor alternative structure. Although we anticipated a multidimensional structure for the ARBS (based on literature related to the JD–R model), it is possible that a single factor may better represent the antecedents of burnout. Formal comparisons indicated that the proposed two-factor structure did, in fact, outperform a single-factor alternative (ΔCFI = 0.18; Δ*χ*^2^(1) = 147.30, *p* < 0.001), supporting the distinct yet related nature of the ARBS subscales.

#### 3.5.2. Measurement Invariance Across Gender

Given established gender differences in emotional intelligence, coping strategies, and perceptions of relational strain (e.g., [73]; [58]), it was important to evaluate whether the structure of the ARBS operates equivalently for men and women. Thus, multigroup CFAs using the MLR estimator supported full configural, metric, and scalar invariance across women and men. Changes in fit were negligible (ΔCFI = 0.003–0.010; ΔRMSEA = 0.000–0.001), and robust Satorra-Bentler difference tests corroborated these conclusions; constraining latent means did not worsen fit (*p* = 0.20). Although absolute fit indices were moderate (CFI ≈ 0.72; RMSEA ≈ 0.10), the invariance pattern supports the equivalence of measurement structure across gender and indicates that the ARBS functions similarly for men and women.

#### 3.5.3. Psychometric Properties

The ARBS subscales showed strong, theory-consistent associations with relational constructs. In fact, both subscales were negatively correlated with relationship satisfaction (*r* = −0.85 and −0.67, *p*s < 0.001) and positively related to stress and relational strain (*r* = 0.21–0.33, *p*s < 0.001) as well as trust and intimacy concerns (*r* = 0.59–0.70, *p*s < 0.001). Hierarchical regressions demonstrated incremental validity beyond general stress when predicting satisfaction (Δ*R*^2^ = 0.12, *p* < 0.001), indicating that the ARBS captures unique variance in relational functioning above and beyond general distress measures.

A latent two-factor SEM (*N* = 288; 36 indicators) further supported predictive validity. Using MLR with FIML estimation, overall fit was acceptable given model complexity (robust CFI ≈ 0.79; robust TLI ≈ 0.79; robust RMSEA ≈ 0.11; SRMR = 0.06). Critically, the Depletion/Incompatibilities factor uniquely predicted lower relationship satisfaction (β = −0.95, *p* < 0.001) and higher infidelity urge (*β* = 0.35, *p* = 0.024), whereas Resource Deficits showed no unique effects once Depletion/Incompatibilities was included (|*β*| ≤ 0.13, *p*s ≥ 0.17). An auxiliary MLR model including the dichotomous therapy variable yielded convergent inferences and indicated that therapy participation was not uniquely predicted by either factor.

Finally, to confirm internal consistency, Cronbach’s alphas were again calculated for both subscales. Both the Relationship Depletion and Exhaustion subscale (*α* = 0.97) and the Relational Overload subscale (*α* = 0.89) demonstrated good scale reliability.

### 3.6. Discussion

The purpose of Study Two was to confirm the factor structure of the ARBS and establish its psychometric properties. Across CFA, invariance testing, correlational and regression analyses, and SEM modeling, the ARBS demonstrates a coherent psychometric profile: a stable two-factor structure that generalizes across gender, strong convergent and incremental validity, and theoretically patterned predictions of relational outcomes. Together, these findings establish the ARBS as a psychometrically rigorous instrument for assessing relationship burnout, elucidating how relational demands and depleted resources contribute to strain, disengagement, and reduced satisfaction in contemporary partnerships.

As expected, ARBS scores were strongly associated with relationship satisfaction. This aligns with research showing that satisfaction in close relationships is highly sensitive to unmet needs and unmanaged demands ([59]; [49]; [32]). However, the ARBS did not predict therapy participation. Given that therapy participation was measured dichotomously, the lack of association may partly reflect restricted variance. Future research could employ more nuanced indicators (e.g., frequency, duration, or type of therapy) and examine these associations within samples that are more likely to exhibit meaningful variability in depletion and exhaustion. Specifically, a clinical population (e.g., couples currently in therapy) or individuals with prior therapy experience represent the most logical groups for evaluating whether ARBS scores correspond to treatment-seeking behavior, as these populations are more likely to experience elevated relational strain and thus provide the necessary range to test these associations robustly.

One explanation is that help-seeking decisions are not purely driven by relational strain, but by logistical and attitudinal barriers such as financial cost, stigma, partner willingness, or cultural norms about therapy. In fact, research indicates that there are many factors that influence participation in therapy, highlighting that barriers such as cost, scheduling difficulties, lack of trust in therapists, uncertainty about the therapeutic process, and relational dynamics ([33]). Finally, Relationship Depletion and Exhaustion predicted the urge to commit infidelity, consistent with both the investment model ([62]; [21]) and infidelity-specific frameworks ([28]). Proponents of these models posit that when intimacy, appreciation, and responsiveness decline, individuals perceive lower rewards and investment in the relationship and become more open to alternative partners.

## 4. General Discussion

The present research sought to develop and validate the Antecedents of Relationship Burnout Scale (ARBS), a psychometrically rigorous tool designed to assess the mechanisms that give rise to relationship burnout (RB). Prior instruments, such as the Couple Burnout Scale, have been limited by vague and symptom-focused items that overlook the underlying causes of relational strain and privilege monogamous, dyadic contexts. In response, the ARBS was developed as the field’s first multidimensional and theoretically grounded measure of RB, guided by the JD–R framework ([6]; [16]). By distinguishing between relational demands that drain partners’ capacities and the erosion of relational resources that sustain connection, the ARBS advances a comprehensive understanding of how burnout unfolds within intimate partnerships.

Findings from both studies indicated that a two-factor structure of the ARBS was ideal, comprising Relational Overload and Relational Depletion and Exhaustion. Together, these factors reflect two interrelated yet distinguishable dimensions of relational strain: the accumulation of excessive relational demands and the gradual erosion of emotional, cognitive, and motivational resources ([6]; [16]; [59]). This structure provides a coherent framework for characterizing how individuals experience relational strain and offers a more nuanced lens through which to assess the multifaceted nature of relational burnout.

Both ARBS dimensions were significantly associated with lower relationship satisfaction, and the Relational Depletion and Exhaustion factor was further linked to a stronger urge to engage in infidelity. These associations confirm the theoretical validity of the scale, indicating that RB involves both emotional exhaustion and reduced relational engagement, consistent with conceptualizations of burnout as a chronic resource imbalance ([45]).

Importantly, the observed associations between the ARBS factors and relationship quality indicators suggest that RB reflects the functioning of the relational system as a whole, rather than solely individual experiences. Although the studies were conducted at the individual level, the conceptual foundation of ARBS aligns with contemporary perspectives in relationship science that emphasize interdependence and co-regulatory processes within interpersonal systems ([12]; [58]). Within this framework, relational demands and resources are co-created and mutually regulated in daily interactions, making stress and recovery inherently relational rather than purely individual phenomena. Accordingly, the ARBS offers a foundation for future studies that may extend to other forms of close bonds (e.g., friendships, caregiving, or non-mononormative relationships) through which sustained emotional engagement and resource imbalance can similarly emerge.

The findings also contribute to the growing body of research adapting the JD–R model to non-work domains. In education, the Study Demands–Resources model ([40]) has demonstrated analogous links between excessive demands and reduced engagement. The current research extends this trajectory by demonstrating that the JD–R logic can transcend domains, effectively describing processes of overload and resource loss within close relationships. In this sense, the ARBS fills a theoretical and methodological gap, providing the first empirically grounded tool for studying antecedents of RB.

### Limitations and Future Directions

Despite robust psychometric support, several limitations should be noted. First, the cross-sectional and self-report nature of the data limits causal inference about the directionality of effects between overload, depletion, and relationship quality. This design also relied exclusively on individual-level reports, which prevents us from examining how relational burnout is experienced and expressed within the dyad. To address these concerns, future research should adopt longitudinal designs to capture the temporal dynamics of relational burnout and recovery (i.e., whether burnout contributes to lower relationship quality or whether low-quality relationships give rise to burnout). In addition, a dyadic approach represents an important next step, as it would allow researchers to assess concordance and divergence in partners’ reports of burnout and explore how each person’s strain may shape both their own outcomes and their partner’s experiences ([35]).

Second, although the samples were inclusive in design, participants were predominantly from Western, monogamous contexts, restricting generalizability. Future research should attempt the ARBS structure across diverse cultural and structural configurations (e.g., open relationships, polyamory). Third, while the ARBS focuses on antecedents of burnout, it does not directly assess its affective manifestations (e.g., emotional distancing) or physiological symptoms of RB. Thus, future studies should adopt multimethod approaches, combining self-reports with behavioral and physiological data (e.g., cortisol synchrony, heart-rate variability) to capture additional nuance in the experience and expression of RB, including how relational, emotional, and biological indicators co-occur and fluctuate across time and context.

## 5. Conclusions

In conclusion, this research redefines RB as a dynamic imbalance between demands and resources, rather than a terminal emotional state. The ARBS offers a theoretically grounded and empirically validated measure for examining this process in relational contexts, contributing to a more nuanced understanding of how chronic emotional investment without adequate recovery leads to exhaustion and disengagement. In doing so, it advances both the study of relational burnout and the evolution of the JD–R model itself. Thus, the ARBS represents a significant step toward integrating theories of stress, emotional regulation, and well-being within the broader science of relationships.

## Figures and Tables

**Figure 1 behavsci-15-01737-f001:**
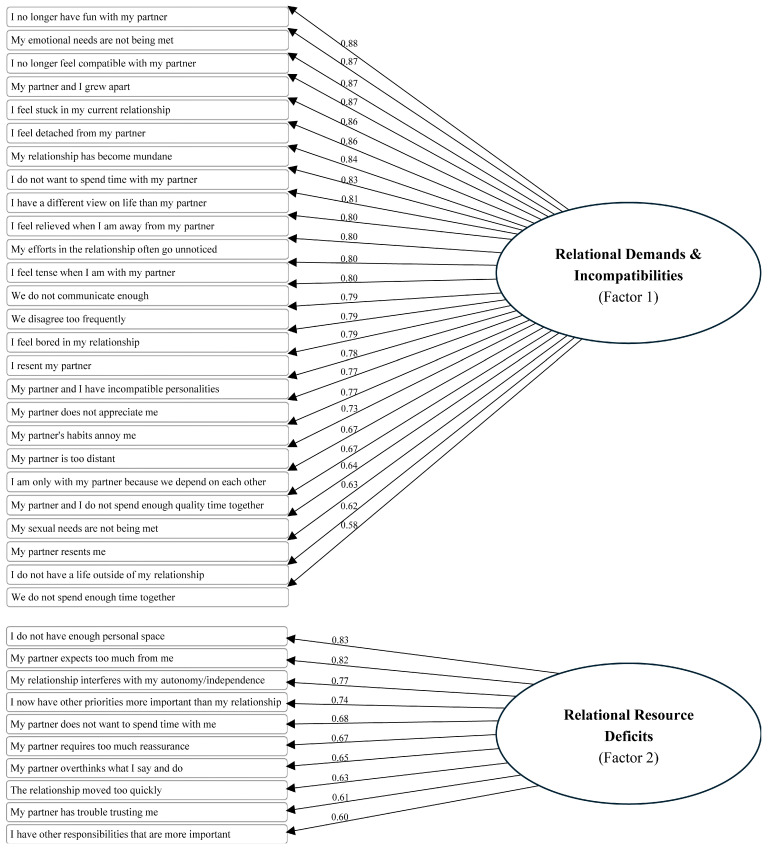
Path diagram and factor loadings obtained from CFA.

**Table 1 behavsci-15-01737-t001:** Factor Loadings for the Antecedents for Relationship Burnout Scale (ARBS).

	Factor 1	Factor 2	*h* ^2^
I feel detached from my partner	**0.96**	−0.15	0.79
I no longer feel compatible with my partner	**0.95**	−0.07	0.83
My partner does not appreciate me	**0.91**	−0.09	0.71
I feel bored in my relationship	**0.89**	−0.04	0.74
We do not communicate enough	**0.87**	−0.04	0.72
My partner is too distant	**0.86**	0.01	0.75
I feel stuck in my current relationship	**0.84**	−0.01	0.75
I resent my partner	**0.84**	0.04	0.76
I no longer have fun with my partner	**0.83**	0.02	0.70
My emotional needs are not being met	**0.83**	−0.02	0.63
My sexual needs are not being met	**0.82**	−0.08	0.55
I feel relieved when I am away from my partner	**0.82**	0.08	0.79
My partner does not want to spend time with me	**0.79**	0.07	0.70
My relationship has become mundane	**0.78**	0.04	0.65
My partner and I grew apart	**0.72**	0.02	0.61
I have a reduced desire for physical affection	**0.72**	0.13	0.66
My partner’s habits annoy me	**0.69**	0.07	0.55
My partner has a reduced desire for physical affection	**0.68**	0.02	0.51
I feel tense when I am with my partner	**0.68**	0.19	0.68
I do not want to spend time with my partner	**0.61**	0.18	0.58
We disagree too frequently	**0.59**	0.26	0.68
My efforts in the relationship often go unnoticed	**0.58**	0.29	0.66
I have trouble trusting my partner	**0.58**	0.18	0.50
My partner and I have incompatible personalities	**0.56**	0.28	0.61
My partner is attracted to other people	**0.53**	0.22	0.61
I am only with my partner because we depend on each other	**0.52**	0.24	0.59
I have a different view on life than my partner	**0.50**	0.31	0.57
We do not spend enough time together	**0.48**	0.27	0.75
My partner resents me	**0.45**	0.20	0.41
I do not have a life outside of my relationship	**0.44**	0.33	0.52
My partner and I do not spend enough quality time together	**0.42**	0.32	0.49
I am attracted to other people	**0.37**	0.29	0.44
My partner requires too much reassurance	−0.30	**0.98**	0.60
I have other responsibilities that are more important	−0.14	**0.80**	0.46
I am experiencing extended family tension/concerns	0.00	**0.76**	0.61
My partner expects too much from me	0.02	**0.75**	0.62
My partner overthinks what I say and do	0.09	**0.74**	0.66
I have previous relationship trauma that impacts my current relationship	−0.02	**0.72**	0.48
I overthink what my partner says and does	0.10	**0.66**	0.54
My partner has previous relationship trauma that impacts our current relationship	0.06	**0.66**	0.52
My partner has trouble trusting me	0.15	**0.64**	0.58
My partner has other responsibilities that are more important	0.06	**0.57**	0.37
The relationship moved too quickly	0.20	**0.54**	0.49
I now have other priorities more important than my relationship	0.28	**0.53**	0.59
My relationship interferes with my autonomy/independence	0.33	**0.50**	0.59
I am spending more energy on the relationship than my partner	0.24	**0.48**	0.45
I do not have enough personal space	0.33	**0.46**	0.53
My partner and I are experiencing infertility issues	0.29	**0.37**	0.39

Note. Only loadings ≥ 0.35 were interpreted. Factor 1 relates to strain, incompatibility, loss of autonomy, resentment, and relational misfit (Relational Demands and Incompatibilities). Factor 2 relates to not feeling valued/supported, lack of communication and intimacy, unmet emotional/sexual needs, and partner distance (Relational Resource Deficits). *h*^2^ = the extracted communality. Bolded factor loadings indicate the factor to which each item was assigned based on its highest loading.

**Table 2 behavsci-15-01737-t002:** Means and Standard Deviations for the Antecedents for Relationship Burnout Scale (ARBS).

	Study One		Study Two	
	*Mean*	*SD*	*Mean*	*SD*
I have other responsibilities that are more important	2.72	1.69	2.34	1.74
My partner and I do not spend enough quality time together	2.62	1.72	2.98	2.12
I overthink what my partner says and does	2.48	1.65	-	-
My partner has other responsibilities that are more important	2.48	1.61	1.85	1.47
My partner expects too much from me	2.43	1.72	2.26	1.89
My partner requires too much reassurance	2.43	1.66	1.97	1.66
I am spending more energy on the relationship than my partner	2.43	1.74	-	-
I have a different view on life than my partner	2.43	1.64	2.29	1.88
We do not spend enough time together	2.38	1.71	2.53	2.07
My partner’s habits annoy me	2.36	1.59	2.55	1.78
My financial needs are not being met	2.31	1.72	-	-
I have previous relationship trauma that impacts my current relationship	2.30	1.63	-	-
My sexual needs are not being met	2.29	1.75	2.02	1.66
My partner and I grew apart	2.28	1.72	2.41	1.98
I am attracted to other people	2.27	1.72	-	-
My partner overthinks what I say and do	2.27	1.60	2.12	1.89
We do not communicate enough	2.26	1.73	2.38	1.93
We disagree too frequently	2.25	1.55	1.67	1.31
My emotional needs are not being met	2.22	1.63	1.91	1.53
I now have other priorities more important than my relationship	2.19	1.64	2.11	1.51
My efforts in the relationship often go unnoticed	2.17	1.65	2.26	1.58
The relationship moved too quickly	2.15	1.64	1.85	1.31
I have a reduced desire for physical affection	2.13	1.53	-	-
My partner does not appreciate me	2.12	1.65	2.12	1.88
I no longer feel valued by my partner	2.10	1.59	-	-
My partner has a reduced desire for physical affection	2.09	1.60	-	-
I do not have a life outside of my relationship	2.06	1.58	1.96	1.89
My relationship interferes with my autonomy/independence	2.06	1.44	1.86	1.63
I no longer have fun with my partner	2.06	1.52	2.09	1.74
My partner resents me	2.05	1.65	1.85	1.51
My partner has previous relationship trauma that impacts our current relationship	2.05	1.54	-	-
My partner and I have incompatible personalities	2.04	1.59	2.41	2.04
I feel stuck in my current relationship	2.03	1.57	1.97	1.63
I am only with my partner because we depend on each other	2.02	1.55	1.96	1.74
I do not have enough personal space	2.02	1.60	2.26	1.84
I feel detached from my partner	2.00	1.54	2.04	1.76
I feel bored in my relationship	2.00	1.55	1.98	1.63
My partner is too distant	1.99	1.51	1.95	1.61
My relationship has become mundane	1.97	1.44	1.96	1.51
I feel relieved when I am away from my partner	1.96	1.58	2.59	2.01
My partner is attracted to other people	1.96	1.50	-	-
I no longer feel compatible with my partner	1.96	1.53	2.12	1.53
I have trouble trusting my partner	1.95	1.54	-	-
I feel tense when I am with my partner	1.94	1.41	1.93	1.55
I am experiencing extended family tension/concerns	1.93	1.37	-	-
My partner has trouble trusting me	1.91	1.46	2.37	1.95
I do not want to spend time with my partner	1.90	1.45	1.86	1.50
My partner does not support my decisions	1.87	1.39	-	-
I resent my partner	1.78	1.34	1.71	1.51
My partner does not want to spend time with me	1.74	1.23	1.74	1.45
My partner and I are experiencing infertility issues	1.49	1.14	-	-

## Data Availability

The data presented in this study are openly available in [OSF] [https://osf.io/9cner/overview?view_only=9879244dbf8f4a4b8bd6ad3d33d3990b].

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
