# Peer review of "Love on Empty: The Development and Validation of a Comprehensive Scale to Measure Burnout in Modern Relationships"

_behavsci, 2025, doi:10.3390/bs15121737_

Round 1

Reviewer 1 Report

Comments and Suggestions for Authors

Article is outstanding and is a contribution to the field. The final scale should be included at the end of the article.

Author Response

REVIEWER ONE

Comment 1: The final scale should be included at the end of the article.

Response 1: Thank you for the suggestion. We have now included the final draft of the scale in Appendix 1, which will be included at the end of the paper.

Reviewer 2 Report

Comments and Suggestions for Authors

Author Response

REVIEWER TWO

Comment 2: Line 62: Avoid starting paragraphs with demonstrative pronouns (“this”) as there is nothing to be referred to by this term. “The” would work just as well here.

Response 2: Thank you. We have now revised the sentence to start with “The” rather than “This.”

Comment 3: Lines 62-66: These sentences require citations to support the claims being made.

Response 3: Thank you. In-text citations and references have now been included.

Comment 4: Line 70: Could Freudenberger be cited here?

Response 4: Of course. This has now been added.

Comment 5: Line 74: The citation should read “(World Health Organization, 2022)” and the ICD-11 can go in-text (I will defer to the editor on whether this should be the full term or if the abbreviation is fine on first use).

Response 5: Thank you, this has been revised accordingly.

Comment 6: Line 75: Similar demonstrative pronoun issue here (“Building on this framework”).

Response 6: This has now been reworded.

Comment 7: Line 79: I suggest removing the definitional content from parentheses as it is essential to reader understanding, not supplemental. 

Response 7: The definition has now been removed from parentheses.

Comment 8: Line 81: “Most studies” is more concise expression here.

Response 8: Thank you. See revision on page 2.

Comment 9: Line 85: Missing a sentence here that explains that infertility and infidelity are not the only determinants of RB.

Response 9: We have now clarified that other predictors have been assessed on line 83.

Comment 10: Line 92-95: Citations could be added here.

Response 10: Citations have now been included on lines 94 and 97.

Comment 11: Line 102: Pines (2011) should be formatted as a formal citation. Furthermore, the authors should format this as a narrative citation so that it applies to the following sentences that draw from that source.

Response 11: A narrative citation has now been used on line 105.

Comment 12: Line 105: While I agree that these items could be directly unrelated to the relationship, they could still impact the relationship (burnout) and therefore should not be dismissed so readily.

Response 12: We agree. Thus, we reworded the sentences on lines 107 and 108 to make clear that we are critiquing the lack of validation employed in Pines’ work.

Comment 13: Line 119: This is quite a long paragraph and could be made more concise/contain fewer sentences through merging/editing sentences.

Response 13: We thank the reviewer for this helpful suggestion. In response, we removed some of the language in question and restructured the content into three shorter paragraphs. We hope these revisions improve clarity and readability.

Comment 14: Line 161: I am not sure I understand why inadequate resources would have a “protective role” in the context of RB?

Response 14: The was a typo on our part. We have now corrected this on line 166.

Comment 15: Line 162: It would be more appropriate to state that you explored the factor structure or whether the bi-dimensional structure was appropriate because you performed an EFA.

Response 15: Thank you. This has been reworded in line 168.

Comment 16: Line 185: I believe the clarification of “unmarried” is required ahead of monogamous relationship is required, otherwise, it is unclear what the distinction is between these subsamples.

Response 16: Great point. To clarify, “unmarried” was now added to line 191.

Comment 17: Line 192: There is a need for more clarity around how these online platforms were used. It appears that the platforms were used to actively collect data (i.e., researchers asked questions), but many studies use the platforms to passive collect data (i.e., researchers collect pre-existing data).

Response 17: The use of Prolific to actively collect all data was clarified in lines 219 and 222.

Comment 18: Line 200: What was the nature of this undergraduate pilot sample? In a relationship as the main sample was? Age, gender identity, sexual identity? This is relevant information if the authors have it.

Response 18: Details pertaining to the undergraduate pilot sample have now been included in lines 206-201.

Comment 19: Line 240: These items see conceptually relevant to RB, was no attempt made to keep them?

Response 19: Although they were relevant, we consulted the team it was unanimous that these items be removed for scale clarity.

Comment 20: There are only 33 items in this table, but the text would suggest there should be 49 items. Furthermore, the text suggests there are 32 items in the first factor, so why are there 33 here? I believe the items loading onto the second factor have not been included. This needs to be amended or clarified.

Response 20: We apologize, it does not look like Table 1B was included in the submission. We have now aggregated both factors into Table 1.

Comment 21: Some factor loadings here are extremely high, which could indicate multicollinearity. I understand that the authors planned a further study, but there should have been work done at this point to clarify whether items were multicollinear/redundant. The authors should be reporting and/or interpreting their Determinant value and communalities at a minimum here, as if this EFA was the basis for further analyses/studies, there could be flaws that are carried battleforward.

Response 21: We thank you for pointing this out. We have no included their communalities in Table 1.

Comment 22: Line 257: As above, this alpha value is extremely high and could be due to multicollinearity. The model should have been refined further before progressing.

Response 22: Although the two factors were strongly correlated, we opted to wait to conduct a more thorough review of item redundancy until the CFA, as this approach provides a more rigorous, theory-driven test of the measurement structure and allows for targeted item reduction using model-fit indices and modification patterns.

Comment 23: Line 293: Remove demonstrative pronoun to start paragraph.

Response 23: Thank you, this has now been completed.

Comment 24: Line 303: In the context of the above issues with the EFA, these conclusions about the applicability of the JD-R feel a bit hasty, but the points are logical and likely valid.

Response 24: Good point. We have now tempered our language to present the findings more cautiously.

Comment 25: Line 426: I do wonder if some of these items that were removed during CFA could have been removed with the additional EFA steps/models described above. If this was the case, it would be sounder that they are removed during EFA than CFA in the context of this paper.

Response 25: We politely disagree. We opted to retain as many items as possible during the EFA stage in an effort to rely on the model-fit indices and modification patterns provided by the CFA.

Comment 26: Line 481: This Cronbach’s alpha is still extremely high, which makes me wonder if there is scope to remove even more redundant items.

Response 26: Although we agree, we ultimately avoided omitting additional items given that 10 items had already been removed during the refinement process and preliminary floor effects were evident (means for the ARBS items were relatively low). To address this concern, we added a limitation and future direction, noting the need to recruit samples characterized by higher levels of relational distress or exhaustion, as greater variability in symptom severity would allow for a more accurate evaluation of item performance (lines 590–598).

Comment 27: Line 497: The logical sample to test these alternative measures on would be a clinical (in-therapy) population and/or those who have previously used therapy.  This point also applies to the next paragraph.

Response 27: We completely agree. This next step has been described in lines 516-523 and again in the limitations/future directions section.

Comment 28: Line 529: The directional claim based on a correlation here is unwarranted. It is not clear how this study could make this type of claim, and it should be revised or removed.

Response 28: Thank you, we have omitted this sentence and revised the paragraph accordingly.

Comment 29: Line 572: I expected to see a suggestion about dyadic study design/data collection here. I think this is a logical and warranted suggestion based on your points about the individualised nature of your data.

Response 29: Good call. We added a future direction recommending the use of the Actor–Partner Interdependence Model to examine actor and partner effects in relational burnout. This addition appears in lines 582–593.

Reviewer 3 Report

Comments and Suggestions for Authors

Overall this is a very robust tool that was developed for the sake of determining relationship depletion. I do believe that this will have a significant impact on the way practitioners and scientists view relationships in decline. As a new measurement tool is being developed here, I would have liked to see some Crohnbach's Alpha calculations made. I would have also liked to see some more recent work on relationships influencing this research and cited in the references list. However, overall, the research is robust, the graphics and models displayed make sense. I appreciated the creativity that comes with developing a measure based on findings from Reddit, Quora, and other online spaces in which people have complained about their relationships! 

Author Response

REVIEWER THREE

 Comment 30: I would have liked to see some Crohnbach's Alpha calculations made. I would have also liked to see some more recent work on relationships influencing this research and cited in the references.

Response 30: We apologize if we are overlooking something, but alphas were reported for both subscales in both studies. Should we report them elsewhere in the manuscript? More recent references have now been included in the text and in the reference section.

Round 2

Reviewer 2 Report

Comments and Suggestions for Authors

The authors have done a good job attending to the issues identified in my first review. While I would personally prefer further EFA prior to CFA, the justification provided by the authors is sufficient and the extracted communalities are encouraging. However, I believe there are two revisions that are essential before publication.

  1. The factor loading values have changed in Table 1. I am not sure why this is the case, but it introduces a major issue with a factor loading value of 1.01 for the first item listed. This is not possible and indicates an issue. Some options for the authors here are:
    1. Re-run their EFA to resolve this issue (e.g., identify the likely highly correlated item[s] that result in this value) 
    2. Place a note on the problematic value in the table that acknowledges the nature of it, and indicates that the values are presented as is because the planned CFA was the remedy opted for.
  2. The above should be complemented with some writing alongside the justifications provided starting on line 290. In the current manuscript, the authors' decisions are justified but remain open to valid statistical and methodological critique. The authors must communicate their understanding of the issues present in the EFA and strength their decision to push forward with the CFA.

I am selecting reconsider after major revision because I believe this is an essential change before publication, the manuscript is otherwise publishable.

Author Response

Comment 1: The factor loading values have changed in Table 1. I am not sure why this is the case, but it introduces a major issue with a factor loading value of 1.01 for the first item listed. This is not possible and indicates an issue. Some options for the authors here are:

  1. Re-run their EFA to resolve this issue (e.g., identify the likely highly correlated item[s] that result in this value) 
  2. Place a note on the problematic value in the table that acknowledges the nature of it, and indicates that the values are presented as is because the planned CFA was the remedy opted for.

Response 1: Thank you for this helpful observation. When reconducting the EFA to report communalities, we discovered that several factor loadings had been incorrectly reported in the initial version of the manuscript. These errors were corrected during revision. Regarding the factor loading that exceeded 1.0, this issue has now been resolved. Upon further inspection, we identified a second item (“I no longer feel valued by my partner”) that was highly correlated with the problematic item. This newly identified redundancy produced the problematic loading of 1.01. To address this, the redundant item was removed prior to running the final EFA and all Study 2 analyses. We have now added justification for the removal of this item in lines 205–253, and all subsequent statistics in both studies have been updated to reflect this adjustment.

Comment 2: The above should be complemented with some writing alongside the justifications provided starting on line 290. In the current manuscript, the authors' decisions are justified but remain open to valid statistical and methodological critique. The authors must communicate their understanding of the issues present in the EFA and strength their decision to push forward with the CFA.

Response 2: Thank you for this suggestion. In response to your feedback, we have strengthened our discussion of the EFA limitations and clarified our rationale for proceeding with the CFA. Specifically, we expanded the writing in the section beginning on line 293 to explicitly acknowledge the residual item redundancies and justify why advancing to a CFA remained appropriate. As noted, CFA offers a more stringent and theoretically guided test of model fit and is commonly used as the next step even when EFA results reveal signs of item overlap. We now articulate this rationale more clearly to demonstrate our understanding of the limitations and reinforce the methodological soundness of our decision to move forward with the CFA.

Round 3

Reviewer 2 Report

Comments and Suggestions for Authors

It is great to hear that the problematic item inter-correlation was identified and managed with further analyses. I note that Table 1 still shows the factor loading value of 1.01, so it is unclear if the removal of the item actually resolved the problematic loading. As the problematic item has been deleted in Table 1, I assume that this table is now reporting the values post-item removal. There is potential for all of the values to have changed slightly after the removal of that item, so I recommend all Table 1 values are reviewed and amended as necessary prior to final publication. Alternatively, the reporting of the old values could be made transparent in-text and in the Table 1 information.

The content added to further explain the nature of the EFA results and justification to proceed with CFA is well done and valuable. 

Author Response

Comment 1: It is great to hear that the problematic item inter-correlation was identified and managed with further analyses. I note that Table 1 still shows the factor loading value of 1.01, so it is unclear if the removal of the item actually resolved the problematic loading. As the problematic item has been deleted in Table 1, I assume that this table is now reporting the values post-item removal. There is potential for all of the values to have changed slightly after the removal of that item, so I recommend all Table 1 values are reviewed and amended as necessary prior to final publication. Alternatively, the reporting of the old values could be made transparent in-text and in the Table 1 information.

The content added to further explain the nature of the EFA results and justification to proceed with CFA is well done and valuable. 

Response 1: Our apologies, the factor loading was never revised in the table. It has now been corrected. Thank you for catching our oversight.